# Calcitriol Promotes Differentiation of Glioma Stem-Like Cells and Increases Their Susceptibility to Temozolomide

**DOI:** 10.3390/cancers13143577

**Published:** 2021-07-16

**Authors:** Julia Gerstmeier, Anna-Lena Possmayer, Süleyman Bozkurt, Marina E. Hoffmann, Ivan Dikic, Christel Herold-Mende, Michael C. Burger, Christian Münch, Donat Kögel, Benedikt Linder

**Affiliations:** 1Neuroscience Center, Experimental Neurosurgery, Department of Neurosurgery, Goethe University, 60590 Frankfurt am Main, Germany; Julia.gerstmeier@outlook.de (J.G.); annalena.possmayer@gmail.com (A.-L.P.); koegel@em.uni-frankfurt.de (D.K.); 2Faculty of Medicine, Institute of Biochemistry II, Goethe University Frankfurt, 60590 Frankfurt am Main, Germany; Bozkurt@med.uni-frankfurt.de (S.B.); Hoffmann@med.uni-frankfurt.de (M.E.H.); dikic@biochem2.uni-frankfurt.de (I.D.); ch.muench@em.uni-frankfurt.de (C.M.); 3Division of Experimental Neurosurgery, Department of Neurosurgery, University Hospital Heidelberg, INF400, 69120 Heidelberg, Germany; Christel.Herold-Mende@med.uni-heidelberg.de; 4Dr. Senckenberg Institute of Neurooncology, Goethe University Hospital, 60528 Frankfurt am Main, Germany; michael.burger@kgu.de; 5German Cancer Consortium DKTK Partner Site Frankfurt/Main, 60590 Frankfurt am Main, Germany; 6German Cancer Research Center DKFZ, 69120 Heidelberg, Germany

**Keywords:** calcitriol, vitamin D_3_, glioblastoma, glioblastoma stem-like cells

## Abstract

**Simple Summary:**

Cancer cells with a stem-like phenotype that are thought to be highly tumorigenic are commonly described in glioblastoma, the most common primary adult brain cancer. This phenotype comprises high self-renewal capacity and resistance against chemotherapy and radiation therapy, thereby promoting tumor progression and disease relapse. Here, we show that calcitriol, the hormonally active form of the “sun hormone” vitamin D_3_, effectively suppresses stemness properties in glioblastoma stem-like cells (GSCs), supporting the hypothesis that calcitriol sensitizes them to additional chemotherapy. Indeed, a physiological organotypic brain slice model was used to monitor tumor growth of GSCs, and the effectiveness of combined treatment with temozolomide, the current standard-of-care, and calcitriol was proven. These findings indicate that further research on applying calcitriol, a well-known and safe drug, as a potential adjuvant therapy for glioblastoma is both justified and necessary.

**Abstract:**

Glioblastoma (GBM) is the most common and most aggressive primary brain tumor, with a very high rate of recurrence and a median survival of 15 months after diagnosis. Abundant evidence suggests that a certain sub-population of cancer cells harbors a stem-like phenotype and is likely responsible for disease recurrence, treatment resistance and potentially even for the infiltrative growth of GBM. GBM incidence has been negatively correlated with the serum levels of 25-hydroxy-vitamin D_3_, while the low pH within tumors has been shown to promote the expression of the vitamin D_3_-degrading enzyme 24-hydroxylase, encoded by the *CYP24A1* gene. Therefore, we hypothesized that calcitriol can specifically target stem-like glioblastoma cells and induce their differentiation. Here, we show, using in vitro limiting dilution assays, quantitative real-time PCR, quantitative proteomics and ex vivo adult organotypic brain slice transplantation cultures, that therapeutic doses of calcitriol, the hormonally active form of vitamin D_3_, reduce stemness to varying extents in a panel of investigated GSC lines, and that it effectively hinders tumor growth of responding GSCs ex vivo. We further show that calcitriol synergizes with Temozolomide ex vivo to completely eliminate some GSC tumors. These findings indicate that calcitriol carries potential as an adjuvant therapy for a subgroup of GBM patients and should be analyzed in more detail in follow-up studies.

## 1. Introduction

Gliomas are tumors of the central nervous system, of which the highest grade, glioblastoma (GBM, WHO grade IV glioma), is the most common and the most aggressive form [1]. These tumors are particularly resistant towards conventional chemotherapy, currently consisting of Temozolomide (TMZ)-based radiochemotherapy, with the recent addition of tumor-treating fields (TTF) [2,3,4,5]. The sensitivity of patients towards a TMZ-based therapy is dependent on the presence or absence of the enzyme O^6^-methylguanine (O^6^-MeG)-DNA methyltransferase (MGMT), and many GBMs have a methylated *MGMT*-promoter-region and therefore do not express this enzyme [6]. The MGMT enzyme can repair the DNA damages induced by TMZ. A surgical resection of GBM is often not entirely possible due to highly infiltrative and diffuse growth of these tumors, which routinely leaves some undetectable tumor cells that cause frequent recurrences. All of those factors cause a devastatingly low 5-year survival-rate of less than 5% and a median survival of 15 months after diagnosis [7]. One mechanism by which treatment resistance and tumor recurrence is facilitated is the presence of a sub-population of cancer cells with a stem-like phenotype (GSCs, glioma stem-like cells; reviewed in [8]), which may be induced by several cues such as hypoxia [9], perivascular niches [10] or treatment borders, such as the resection margin [11]. Key characteristics of the cells are that they have a higher differentiation potential, express various marker proteins associated with stemness such as SOX2, SOX9, OLIG2 and are considered more tumorigenic upon transplantation into rodents compared to their differentiated counterparts (reviewed in [8]).

Vitamin D_3_ (VitD_3_) is a steroidal hormone that is synthesized in the skin upon UV-B irradiation, which can alternatively be obtained through dietary sources, such as fatty fish or certain mushrooms [12]. In order to be activated, VitD_3_ has to pass through two hydroxylation steps. The first hydroxylation occurs mainly in the liver to form 25-hydroxy vitamin D_3_ (25(OH)D_3_). This step is mediated by a 25-hydroxylase, CYP2R1. CYP2R1 is the main 25-hydroxylase and it promotes VitD_3_ conversion in the liver, other enzymes with 25-hydroxylase activity, such as CYP27A1, exist in various other organs and can allow for local conversion [13]. The second hydroxylation occurs in the kidney and is mediated by 1α-hydroxylase (encoded by the gene *CYP27B1*) to form the hormonally active form 1α,25(OH)_2_ vitamin D_3_ (1α,25(OH)_2_D_3_), also known as calcitriol. Similarly, calcitriol synthesis can occur in multiple extra-renal sites, including the brain [13]. 25(OH)D_3_ is the major circulating form of VitD_3_ and is frequently employed as a surrogate marker for the VitD_3_ status. The degradation of 25(OH)D_3_ and calcitriol is catalyzed by the 24-hydroxylase (encoded by the gene CYP24A1); the amount of calcitriol is therefore limited by both calcitriol catabolism and decreased amounts of 25(OH)D_3_ available for calcitriol synthesis [14,15]. Calcitriol acts by binding to the vitamin D receptor (VDR), which regulates target gene expression upon translocation to the nucleus. VDR signaling inhibits the expression of genes for calcitriol synthesis (e.g., *CYP27B1*) and activates VDR expression, as well as the expression of genes responsible for calcitriol metabolism such as *CYP24A1* (reviewed in [15]). The activation of genomic VDR signaling also induces anti-tumorigenic effects, such as inhibition of proliferation, stimulation of differentiation processes and apoptosis [16]. In fact, the anti-tumorigenic effects of VitD_3_ are well established, and low 25(OH)D_3_ serum levels are associated with increased cancer risk and mortality, indicating a tumor-preventive function of VDR signaling [15]. Interestingly a recent report demonstrated that acidosis, i.e., the acidic environment around tumors, enhances the stem-like state and mitochondrial respiration of GSCs, which was mediated by a reduction in VitD_3_ via 24-hydroxylase-mediated degradation, which was induced by the low pH [17].

It was recently shown that serum levels of 25(OH)D_3_ and the risk of GBM have an inverse relationship [18], indicating that patients with low 25(OH)D_3_ have a higher risk of developing GBM. Similarly, Zigmont et al. observed that low prediagnostic 25(OH)D_3_ levels are associated with increased risk of glioma in men [19]. In addition, VDR-expression was found to be associated with improved overall survival [20]. Accordingly, calcitriol and/or VitD_3_ have been proposed as potential therapeutics for brain cancers [21,22]. Preclinical studies using differentiated GBM cell lines employed VitD_3_-loaded nanoparticles to induce cytotoxicity in rat glioma cells [23] and combined VitD_3_ with Temozolomide, leading to synergistic cytotoxicity via enhancement of autophagy [24]. Mechanistically, it was shown that calcitriol induces senescence in conventional, FCS-grown glioma cell lines [25]. Importantly, a synthetic derivative of calcitriol, alfacalcidol, has already been tested in a small patient cohort in France as an adjuvant therapy, and it was reported that, among 11 patients analyzed (10 GBMs, 1 Anaplastic Astrocytoma [AA]), 3 patients showed continuous improvement and, finally, complete regression of the tumor (2 GBM, 1 AA); the median survival of all patients was 21 months, with the two GBM responders still alive after 4 years of follow-up [26].

Taken together, these findings lead us to hypothesize that GSCs might be particularly vulnerable for a calcitriol-based therapy. As such, calcitriol should be able to induce differentiation and render GSCs more susceptible to conventional chemotherapeutics, including TMZ.

## 2. Results

### 2.1. Calcitriol Reduces Stemness in GSCs In Vitro

To analyze the potential for a calcitriol-based therapy, we made use of three previously described cell models: NCH644 [27]), GS-5 [28] and the primary culture 17/02 [29]. Using these three cell lines, we performed limiting dilution assays (LDA), as described previously [29,30], and analyzed them using the Web-App ELDA [31]. In principle, the cells were treated using 10, 25 or 50 nM of calcitriol or solvent (Ethanol, EtOH) immediately after seeding. After 7 days, each well was evaluated for the presence of at least one sphere larger than eight cells per well. We observed that calcitriol (Figure 1A–C) significantly reduced the sphere-forming potential (“stem-cell frequency”) of NCH644 (Figure 1A) very strongly and significantly, with a change in the stem-cell frequency from 1 in 23.3 after solvent treatment to 1:81.6 and 1:119.1 after treatment with 10 and 25 nM of calcitriol, respectively. The highest concentration of 50 nM of calctriol induced an almost 10-fold reduction of 1:219.9. The effects on GS-5 (Figure 1B) and 17/02 (Figure 1C) were less pronounced. As such, the stem-cell frequency of GS-5 decreased from 1:188 (EtOH) to 1:206 (10 nM), 1:264 (25 nM) and 1:318 (50 nM), respectively. Only the ~two-fold reduction after treatment with 50 nM calcitriol reached statistical significance. Lastly, the stem-cell frequency of the primary line 17/02 changed from 1:153 after solvent treatment to 1:152 (10 nM), 1:159 (25 nM) to 1:198 (50 nM), and neither was statistically different.

Next, we wondered whether a more clinically applicable drug that is less calcemic than calcitriol, calcipotriol [32], could exert similar effects. Therefore, we also performed LDAs and found, for all three cell lines (Figure 1D–F), similar albeit less intense effects after treatment with 100 nM, 500 nM and 1000 nM of calcipotriol or solvent. Specifically, calcipotriol led to a dose-dependent decrease in the stem-cell frequency of NCH644 cells (Figure 1D) from 1:50.6 (DMSO) to 1:80.5 (100 nM) and 1:104 (500 nM) to 1:129.5 (1000 nM). Only the treatment with 500 nM and 1000 nM led to a statistically significant decrease. For GS-5 (Figure 1E) and the primary line 17/02 (Figure 1F), only the highest concentrations led to a statistically significant decrease in stem-cell frequency. Accordingly, the stem-cell frequency of GS-5 cells decreased from 1:190 (DMSO) to 1:241 (100 nM) to 1:232 (500 nM) and 1:310 (1000 nM), while it changed from 1:158 (DMSO) to 1:192 (100 nM) and 1:209 (500 nM) to 1:232 (1000 nM) in 17/02. The ratios and the respective *p*-values are summarized in Table 1. From these experiments, we drew the conclusion that the synthetic derivative of calcitriol, i.e., calcipotriol, is less potent. In order to describe the therapeutic effects evoked by VitD_3_-signaling, we decided to perform all subsequent experiments with calcitriol only.

During microscopic evaluation of the LDAs, we observed, especially for the well-responding GSC line NCH644-, not only the formation of fewer spheres but the fact that those that did form were markedly smaller. This is exemplified in Figure 2, showing that, compared to solvent treatment (Figure 2A), 10 nM of calcitriol (Figure 2B) and 50 nM of calcitriol (Figure 2C) dose-dependently reduced the amount and size of spheres. This indicates that the analyses using ELDA rather understates the true effect size of calcitriol treatment.

These findings prompted us to analyze the response of the GSCs in more detail. We therefore developed a semi-automatic FIJI-macro to analyze sphere area and number (Figure 3). For NCH644, this approach showed that increasing doses of calcitriol from 10 to 25 and 50 nM significantly reduced the mean sphere number per well (Figure 3A), as well as the median sphere area (Figure 3B). Specifically, the average sphere number and median area of solvent treated cells were 41 and 656 µm^2^, respectively. Treatment with 10 nM, 25 nM or 50 nM reduced these numbers to 24 spheres, with a median size of 510 µm^2^, 27 spheres and with a median size of 506 µm^2^ and 15 spheres with 427 µm^2^, respectively. For GS-5, only the highest dose of 50 nM of calcitriol significantly reduced the number of spheres per well (Figure 3C) compared to solvent treated cells. As such, the mean number of spheres per well was reduced from 71, after solvent treatment, to 58, 71 and 51 after treatment with 10 nM, 25 nM and 50 nM, respectively. The median sphere size (Figure 3D) was 343 µm^2^ after solvent treatment and remained at 329 µm^2^ after treatment with 50 nM of calcitriol. Lastly, the primary culture 17/02 showed neither a reduction in sphere number (Figure 3E) nor a reduction in sphere area (Figure 3F), with an average sphere number and median size of 33 spheres and 420 µm^2^ after solvent treatment and 32 spheres and 414 µm^2^ after 50 nM of calcitriol, respectively.

### 2.2. Differential Activity of Calcitriol

The fact that we observed such pronounced differences between the three GSC lines argued for cell line-/tumor-specific therapeutic effects of calcitriol and prompted us to further expand our cell panel. We therefore analyzed additional 7 GSC lines, including four locally established primary lines. The additional four primary lines are MNOF1244, which has not been described previously (see Materials and Methods), 17/01 [29], MNOF1300 [34] and the primary gliosarcoma line MNOF168 [35]. In addition, we also included the cell lines established and described in the literature, GS-73 [36], NCH421k [27] and NCH481 [37]. We analyzed the sensitivity of the cells using LDAs after treatment with 50 nM of calcitriol for 7 days and calculated the fold-change in stem-cell frequency to better compare the results from the different cell lines. This analysis (Figure 4A) revealed that 6 out of 10 lines (60%) showed a significant reduction in stem-cell frequency after calcitriol treatment (vertical cut-off line in Figure 4A). Importantly, two GSC cultures (NCH644 and NCH421k) show an exceptionally strong response to calcitriol with a more than nine- and seven-fold reduction in stem-cell frequency. Next, we hypothesized that differences in calcitriol-sensitivity might be, at least partially, due to differences in VDR expression, and analyzed VDR mRNA expression via qRT-PCR and plotted it against the calcitriol response (Figure 4B). This approach revealed, on the one hand, that profound differences in VDR expression occurred between the cell lines, ranging over almost three magnitudes from 0.00177 (GS-73) to 1.63088 (NCH644). On the other hand, we determined that a very strong correlation (Pearson R: 0.9828, *p*-value < 0.0001) existed between calcitriol-sensitivity and receptor expression.

### 2.3. Calcitriol Reduces Stemness-Associated Marker Genes

Hereafter, we were interested in the gene expression changes that were induced by calcitriol treatment. Therefore, we treated the cells for 24 and 48 h with 10 and 50 nM of calcitriol, and analyzed target gene expression (Figure 5). Using this approach, we can show that, expectedly, *CYP24A1* expression (Figure 5A) is strongly induced after both 24 and 48 h. Additionally, we analyzed *GLI1* (Figure 5B) as a marker for Hedgehog pathway activity, because this pathway is described to be important for GSC stemness [38], and there is evidence, including from our own previous studies [39,40], that calcitriol is a direct inhibitor of this pathway. Nonetheless, only a small, non-significant tendency towards inhibition can be observed after calcitriol treatment. The analyses of the three stemness marker genes (Figure 5C–E) *OLIG2*, *SOX2* and *SOX9* showed that, after 24 h, *OLIG2* (Figure 5C) expression is significantly reduced after 50 nM of calcitriol treatment and, after 48 h, both 10 and 50 nM treatment. *SOX2* (Figure 5D) is significantly reduced 24 h after treatment with 10 and 50 nM of calcitriol, whereas, after 48 h of treatment, only the 50 nM treatment still elicits a significant reduction. *SOX9* (Figure 5E) only shows a slight tendency towards inhibition after 24 h of treatment, but no change after 48 h. Analyses of the differentiation marker GFAP (Figure 5F) showed a pronounced increase after 24 h, yet only the 10 nM treatment was statistically significant. Moreover, 48 h after treatment, *GFAP*-expression was further increased and statistically significant after treatment with 50 nM of calcitriol. Lastly, we measured *BCL2*-expression (Figure 5G) and *CCND1* (encoding for Cyclin D1) (Figure 5H), which are known to be negatively regulated by active VDR signaling [15]; however, no significant changes were observed, although a slight tendency was apparent for *BCL2*-expression 48 h after treatment with 50 nM of calcitriol.

Using immunofluorescent stainings (Figure 6), we confirmed reductions in SOX2 (Figure 6A,D) and OLIG2 (Figure 6B,E) and an increase in GFAP expression (Figure 6C,F) 72 h after treatment with 50 nM of calcitriol. A quantification (Figure 6D–F) using FIJI showed that the relative signal intensity of SOX2 and OLIG2 decreased to 52% and 53%, respectively, relative to solvent-treated cells (100%). In contrast, GFAP expression increased to 226% compared to solvent, while it should be noted that several cells exhibited a pronounced increase of more than 300%.

### 2.4. Alterations in Global Proteomic Profiles after Calcitriol Treatment

To characterize the effects on global protein levels and to gain a more detailed understanding of the calcitriol effects, we performed quantitative LC-MS/MS. Hierarchal Euclidean clustering analysis showed that replicates of 50 nM of calcitriol treatment (48 h) distinctly differed from solvent and, therefore, they were clustered together (Figure 7A). Then, to determine which proteins were affected the most, we calculated fold differences and significances by two-sided, unpaired Student’s *t*-tests of each protein. Stemness markers such as OLIG2 and SOX2 were significantly reduced, which supports our finding in qPCR and immunofluorescence analysis of these two markers in protein level. It was also determined that most of Ca^2+^-related and VitD_3_-target proteins were significantly upregulated upon calcitriol treatment for 48 h (Figure 7B). Then, we indicated the findings of these proteins with cluster analysis on heatmaps. With the sole exception of AHNAK2, all the altered Ca^2+^-related proteins found, i.e., CAPN5, ANXA2, ANXA5, RCAN1 and S100A6, were increased, suggesting that calcitriol might activate Ca^2+^-related pathways (Figure 7C). CYP24A1 protein levels were significantly elevated (nearly 25-fold), a finding that was strongly correlated with CYP24A1 mRNA levels, as well as other VitD_3_-target proteins; G6PD, LPGAT1, CLMN, ASAP2 and S100A6 were also increased, along with CYP24A1 (Figure 7D). Furthermore, our proteomic analysis showed an elevation in actin/microtubule-related proteins CCT6B, PALLD, ARPC1B, MREG, PACSIN3, AFAP1L2 and TACC2 upon 48 h of calcitriol treatment.

### 2.5. Calcitriol Reduces Tumor Growth Ex Vivo and Synergizes with TMZ

Based on the data obtained so far, we hypothesized that calcitriol has potential as a “differentiation therapy”. This, in turn, would imply that calcitriol-treated cells are more sensitive towards conventional chemotherapy, such as TMZ. To test this hypothesis, we first generated GFP-expressing NCH644 (NCH644 GFP^+^) and NCH421k (NCH421k GFP^+^) cells and applied them in an ex vivo tumor growth assay using organotypic tissue culture (OTC), as described recently [29,41]. For this purpose, the OTCs were prepared and, one day after tumor transplantation (termed d0), they were treated three times per week using solvent (DMSO for TMZ; EtOH for calcitriol), TMZ, calcitriol or a combination of both. For NCH644 (Figure 8), images were taken at days 3, 5, 7 and 10 to measure tumor sizes, with representative images being depicted in Figure 8A. To better compare the dynamics of tumor growth/regression, we normalized the sizes of each tumor to its size at d0. This approach showed that, over time (Figure 8B), solvent-treated tumors (black points/lines) grew continuously. TMZ alone only mildly and non-significantly reduced the tumor areas, which is in line with the concept that GSCs are largely resistant to conventional therapy. In contrast, calcitriol single treatment effectively halted tumor growth of most tumors. Furthermore, other tumors even show complete remission after single-agent calcitriol treatment (8 of 24, 33.33%). Both cases are detailed in Figure 8A. Finally, the combination treatment synergistically and statistically significantly reduces tumor sizes and also leads to the disappearance of several tumors (7 of 27, 25.93%). Specifically, solvent-treated tumors grew to 3.7, 6.0 and 9.8-fold of their initial size after 5 (Figure 8C), 7 (Figure 8D) and 10 days (Figure 8E), while TMZ alone slowed growth non-significantly by 2.8, 3.0 and 5.9-fold. Calcitriol-treated tumors were, on average, 1.1, 1.4 and 2.2-fold larger than their initial size, and the difference was statistically significant for each time point. Importantly, the combination treatment resulted, on average, in 0.8, 0.9 and 1-fold tumor growth compared to initial sizes. This difference was highly significant against solvent and TMZ for all timepoints, as well as against calcitriol-single-treatment after 10 days of treatment, suggesting a potential synergism.

Similar results have been observed for NCH421k GFP^+^ (Figure 9), although these tumors appear to be more sensitive towards TMZ. As such, solvent-treated tumors continuously grew and reached 2.9-, 9.1- and 17.1-fold of their initial size, on average, at day 5 (Figure 9C), day 12 (Figure 9D) and day 16 (Figure 9E), respectively. Treatment with TMZ reduced the growth of the tumors continuously, leading to an average tumor size that was 2.1-(day 5), 0.9-(day 12) and finally 0.4-fold (day 16) compared to the initial tumor. The difference between solvent- and TMZ-treated tumors reached statistical significance after 12 and 16 days. Calcitriol treatment already blocked tumor growth to 0.6-fold after 5 days. This could be further decreased to 0.3-fold (day 12) and finally 0.1-fold (day 16) of the initial tumor, while achieving statistical significance at day 12 and day 16. The combination of both was also highly effective but it did not differ significantly from either treatment. The average tumor size after combination treatment was 1.1-fold (day 5), 0.4-fold (day 12) and 0.3-fold (day 16) compared to initial size. Thus, no synergism can be inferred for NCH421k tumors.

## 3. Discussion

Cancer recurrence generally leads to more aggressive and treatment resistant disease, which finally culminates in the death of patients. Due to highly infiltrative tumor growth, it is impossible to achieve complete microscopic resection in patients with GBM, and it is proposed that the remaining cells can obtain and harbor a stem-like phenotype that is crucial for replenishing the tumor [42,43]. It has been further hypothesized that these GSCs reside in either hypoxic niches [9], perivascular niches [10] or at treatment borders, such as the resection margin [11]. Accordingly, many attempts are being made at specifically targeting this phenotype [44,45,46,47,48]. The anti-cancer and differentiation-promoting activity of VitD_3_ and calcitriol has been known for some time, and it has been shown in numerous studies that intake of VitD_3_, a safe and well-studied drug, can prevent cancer occurrence [15,22,49].

Based on these two paradigms, i.e., its known ability to target the cancer stem-like phenotype and its anti-tumorigenic activity, we decided to apply therapeutic doses of calcitriol, the hormonally active form of VitD_3_, in an aim to reduce the stem-like phenotype of GSCs and thereby increase its sensitivity to chemotherapy. Accordingly, our studies revealed, for the first time, that treatment with 10, 25 and 50 nM of calcitriol decreases stemness in three GSC lines. Notably, the treatment efficiency varied between the cell lines. Some GSCs, for example NCH644, seemed to be more sensitive to calcitriol treatment than GS-5, which showed a moderate response, and 17/02, which barely showed any response at all. Importantly, these findings are in line with previous results obtained in the clinical setting by Trouillas et al. [26]. The authors of this study gave the synthetic VitD_3_-derivative alfacalcidol as an adjuvant therapy to 11 patients (10 GBM, 1 Anaplastic Astrocytoma (AA, grade III glioma)). Of those eleven patients, three (2/11 GBMs, 1/1 AA) experienced continuous improvement and, finally, a complete tumor regression. Accordingly, the median survival of this study cohort was 21 months, which is considerably higher than the reported ~13 months found in the recent literature. The two GBM responders were still alive after 4 years of follow-up [26]. One possible hypothesis derived from these preliminary observations is that a certain sub-population (~20%) of patients may show increased sensitivity towards a VitD_3_-based therapy. This hypothesis is consistent with our analyses of an extended cell panel of 10 GSC cultures. Among those, 60% showed a significant response, whereas 20% (NCH644 and NCH421k) responded strongly to calcitriol both in vitro and ex vivo, providing a solid starting point to experimentally determine molecular factors mediating calcitriol sensitivity in GBM.

Another pertinent question concerns the most effective drug for a VitD_3_-based therapy. Here, we compared the effect obtained by calcitriol to that of its synthetic, less-calcemic [32] derivative, calcipotriol. We observed similar effects in both compounds, although the intensity was comparatively weaker after treatment with calcipotriol, which prompted us to focus our studies on the effects of calcitriol. Another group reported the usage of calcipotriol in T98G serum-grown GBM cells [50]. Similar to our approach, the authors compared calcipotriol to calcitriol and another derivative, tacalcitol, and found that calcitriol is the most effective drug, although they applied very high doses of calcitriol (up to 10 µM). Despite the fact that several studies can be found which employ various VitD_3_-derivatives, such as calcipotriol in pancreatic cancer cells [51,52], comparative approaches are scarce. Using breast cancer cell lines a comparison between calcitriol, calcipotriol and EB1089, another VitD_3_-derivative, was conducted by performing combination treatment with the chemotherapeutic gefitinib [53]. The authors observed synergy of all compounds with the chemotherapy, but no discernable difference was observed between calcitriol and its analogues. Similarly, Colston et al., have shown that both calcitriol and calcipotriol reduce proliferation of MCF-7 breast cancer cells [54]. These findings point towards cancer-type-specific sensitivities.

Based on the functional inhibition of the stem-like phenotype, we next performed gene expression analyses and confirmatory immunofluorescence staining, showing that, aside from the strong and expected [15,55] increase in *CYP24A1*-expression, we observed a highly significant and concentration-dependent reduction in the stemness markers *SOX2*, *OLIG2*, with a concomitant increase in the differentiation marker *GFAP*. These findings indicate that calcitriol indeed reduces the stem-like phenotype of GSCs. This is further corroborated by the lack of changes in *CCND1*-expression, which indicates that the reduced sphere formation, as shown using the LDAs, is not due to inhibition of proliferation of calcitriol, but rather due to prevention of sphere formation via blockage of the stem-like phenotype. We further reasoned that, due to the reduction in stemness, the GSCs might be more prone to conventional chemotherapy, and we therefore wanted to put our hypothesis to test using an advanced, physiological ex vivo approach. Although, we only observed a very moderate trend in the reduction in *BCL2* expression using qPCR, the ex vivo tumor growth assay of NCH644 GSCs showed that TMZ and calcitriol effectively synergized, which could even result in the complete elimination of some tumors. Similarly, Bak et al. observed prolonged survival in TMZ- and VitD_3_-treated rats after orthotopic transplantation, and attributed some of the combined effects to combined, or rather enhanced, activation of autophagy [24], with no changes in apoptosis induction compared to TMZ alone. A sensitization against conventional chemotherapeutics using VitD_3_ or its analogs has also been observed in a variety of in vivo tumor models, such as colorectal cancer [56] or prostate cancer [57]. Conversely, the ex vivo tumor growth assay with NCH421k GSCs showed stronger effects of TMZ alone, and therefore no synergism with calcitriol. NCH644 cells are known to have an unmethylated *MGMT* promoter with a high IC50 of TMZ in vitro (272 mM) [58]; thus, they are rather resistant to TMZ. In contrast, NCH421ks have a methylated *MGMT* promoter and a lower IC50 of TMZ in vitro (200 µM) [58], and are therefore sensitive to TMZ. Thus, these experiments indicate that MGMT-expressing GSCs are particularly amenable to calcitriol-based differentiation therapy, although a larger cohort of GSCs is necessary to foster this hypothesis.

Lastly, we performed a global proteomic analysis of NCH644 GSCs after treatment with calcitriol and could again confirm the reduction in the stemness marker proteins OLIG2 and SOX2, further validating our data and supporting the conclusion that calcitriol induces differentiation of GSCs. In addition, we could confirm that 5 of the 50 upregulated proteins are known VitD_3_-targets, according to Wikipathways (https://www.wikipathways.org/index.php/Pathway:WP2877, [59], accessed on 22 April 2021). Surprisingly, an additional five upregulated proteins are related to Ca^2+^ signaling, indicating an involvement of Ca^2+^ in mediating the calcitriol response. In fact, it is well known that calcitriol can also induce non-genomic responses that involve the release of Ca^2+^ into the cytosol [13,15,60]. One of the upregulated Ca^2+^-related proteins is RCAN1, whose expression is known to be induced via calmodulin–calcineurin-dependent Ca^2+^-signaling [61,62], and VitD_3_ has also been shown to involve calmodulin-dependent signaling [63,64]. These data further indicate that at least a part of the calcitriol-effect is mediated via Ca^2+^ signaling. Following this line-of-thought has been shown that Ca^2+^ is central to the maintenance of neural stem cells, and it is often aberrantly regulated in GSCs [65]. Recently, the term calcium toolbox was proposed for the summary of all Ca^2+^-related proteins in a cell, and it was shown that GBM cells have a different composition of this toolbox compared to healthy brain tissue [66]. In a study by Leclerc et al. [67], the authors compared GSCs grown under stemness conditions to GSCs grown in serum-containing medium for 7 and 30 days, and termed these conditions differentiated and senescent, respectively; they also applied the calcium toolbox created by Robil et al. for a cluster analysis [66]. Interestingly, they found that two of the five Ca^2+^-related proteins from our dataset (ANXA2 and S100A6) were associated with differentiated cells. These findings further corroborate our conclusion that calcitriol induces differentiation and provides a first mechanistic clue through changes in Ca^2+^-signaling or binding proteins. This could also provide a mechanism for the decreases in effectiveness of the less calcemic compound calcipotriol, which only induces genomic VitD_3_-signaling but not the non-genomic axis.

In summary, we provide the first evidence that calcitriol can specifically reduce the stem-like phenotype of a subset of GSCs and potentially synergize with TMZ in MGMT-expressing cells.

## 4. Materials and Methods

### 4.1. Cells and Cell Culture

For the experiments, the following GSC lines were employed: 17/01 [29], 17/02 [29], GS-5 [28], GS-73 [36], the gliosarcoma line MNOF168 [35], MNOF1244 (see below, Section 4.1), MNOF1300 [34], NCH421k [27], NCH481 [37] and NCH644 [27]. All cell lines, except MNOF168, MNFO1244 and MNOF1300, were used and cultured in Neurobasal medium (Gibco, Darmstadt, Germany). The medium was supplemented with 1 × B27, 100 U/mL Penicillin 100 µg/mL Streptomycin (P/S, Gibco), 1 × GlutaMAX (Gibco), 20 ng/mL epidermal growth factor (EGF, Peprotech, Hamburg, Germany) 20 ng/mL and fibroblast growth factor (FGF, Peprotech). The GSCs MNOF168, MNOF1244 and MNOF1300 were grown in DMEM/F12 medium (Lonza, Basel, Switzerland) containing 20 ng/mL each of EGF and FGF, 20% BIT admixture supplement (Pelo Biotech, Planegg/Martinsried, Germany) and P/S. Moreover, 17/01 was a primary cell line derived from a 51-year-old female patient [29]; 17/02 was a primary cell line obtained from a 60-year-old patient after obtaining informed consent and ethics approval [29]. The tumor relapsed after percutaneous radiotherapy (60 Gy), and chemotherapy with TMZ also was performed. The cell line was prepared after a second fractionated irradiation was performed, followed by subtotal resection [29]. GS-5 and GS-73 were gifted by Katrin Lamszus (UKE, Hamburg, Germany). MNOF168 and MNOF1300 were primary cultures provided by Stefan Momma and Julia Tichy (University Hospital Frankfurt, Germany) [34,35]. MNOF1244 was a primary cell culture obtained from a 69-year-old female patient after obtaining informed consent and ethics approval (ethics committee at the University Hospital Frankfurt; reference number 04/09-SNO 01/11). Tumor tissue from the initial surgery prior to radiochemotherapy was dissociated and cultivated under serum-free culture conditions. Christel Herold-Mende (University Hospital Heidelberg, Germany) provided NCH421k, NCH481 and NCH644. HEK293T (ATCC #CRL-3216) was cultured in Dulbecco’s modified Eagle’s medium (DMEM GlutaMAX) supplied with heat-inactivated 10% FBS and P/S (all from Gibco). GFP-positive NCH644 and NCH421k were created by lentiviral transduction (see Section 4.7), as described previously [68].

### 4.2. Compounds

1,25(OH)_2_D_3_ was purchased from Cayman Chemical. Using ethanol (Sigma Aldrich, Taufkirchen, Germany), a stock solution was prepared and stored at −20 °C. Calcipotriol and TMZ were purchased from Sigma-Aldrich. Stock solutions were prepared with DMSO (Carl Roth GmbH) and stored at −20 °C.

### 4.3. Limiting Dilution Assay

The limiting dilution assay was performed as described previously [29,30]. Briefly, 96-well plates were used to seed the cells in 200 µL culture medium. By performing a row-wise descending dilution, cell concentrations of 8, 16, 32, 64, 128, 256, 512 and 1024 cells/well were reached for each cell line, while treatment was performed with calcitriol/calcipotriol, as indicated in the respective figures. The cells were treated immediately after treatment to ensure that single cells are treated and incubated for 7 days. Stem-cell frequencies were assessed 7 days after seeding using extreme limiting dilution analysis (ELDA) software using the standard settings (http://bioinf.wehi.edu.au/software/elda; [31]; accessed on 7 May 2021). After microscopic evaluation, the first two rows per plate were photographed using a Tecan Spark plate reader (Tecan, Grödig, Austria) and used for subsequent analyses. The number and median size of spheres ≥ 250 µm were analyzed by FIJI (v1.52p) [33] using a self-developed macro.

For NCH644 and 17/02, the wells after seeding 512 cells; for GS-5, the wells after seeding 1024 cells were employed for the analysis.

### 4.4. Tagman-Based qRT-PCR

In total, 300.000 cells/well were seeded into 6-well plates and, the following day, were treated with 50 nM calcitriol for 24 h or 48 h. Experiments were performed using 3 biological replicates for each treatment condition, while the experiment was repeated three times. RNA was isolated using the ExtractMe Total RNA Kit (Blirt S.A., Gdanks, Poland), while 1–2 µg RNA was used for cDNA synthesis. SuperScript III System (Life technologies, Darmstadt, Germany) allowed the synthesis of cDNA, whereas 100 U per sample was sufficient. The quantitative Real-Time PCR (qRT-PCR) was performed using Taqman probes (Applied Biosystems, Darmstadt, Geramany), Fast-Start Universal Probe Master Mix (Roche) on a StepOne Plus System (Applied Biosystems) in a 20 µL reaction volume. Ct values were normalized to TATA box-binding protein (TBP). Fold-change in gene expression was determined by 2^−∆∆Ct^ method, except for VDR-expression, whereas the expression was determined using the 2^−∆Ct^ method.

The following Taqman-probes were used: BLC2 (Hs00608023_m1), CCND1 (Hs00765553_m1), CYP24A1 (Hs00167999_m1), GFAP (Hs00909233_m1), GLI1 (Hs00171790_m1), OLIG2 (Hs00300164_s1), SOX2 (Hs01053049_s1), SOX9 (Hs00165814_m1), TBP (Hs00427620_m1), TNC (Hs01115665_m1), VDR (Hs00172113_m1)

### 4.5. Immunofluorescence Staining

8000 cells/well were seeded in 8-well chamber slides and, the following day, were treated for 24, 48, or 72 h with 50 nM calcitriol or solvent. Slides were fixated using 4% PFA before staining with the following primary antibodies: OLIG2 (R&D Systems, Minneapolis, MN, USA), GFAP (Dako Cytomation, Glostrup, Denmark) SOX2 (R&D Systems, Minneapolis, MN, USA) over night. Subsequently, slides were stained with a species-corresponding secondary antibody (goat anti-mouse (Alexa Fluor 594 F(ab’)2 fragment IgG (H+L)); (goat anti-rabbit (Alexa Fluor 488 F(ab’)2 fragment goat anti-rabbit IgG (H+L)); donkey anti-goat (Alexa Fluor 488 IgG (H+L))). Pictures were taken using a Nikon Eclipse TE2000-S fluorescent microscope operated by NIS elements software and adjusted using FIJI. Quantification was performed in Fiji by measuring the integrated density of the fluorescent signal (mean intensity of measured cell multiplied with the measured cell area) of 50 cells per condition using at least three pictures. The area was selected using the freehand selection tool, ensuring an exact and individual measurement for each cell. Violin plots were generated using GraphPad Prism 9.

### 4.6. Proteomic Sample Preparation and Data Analyses

#### 4.6.1. Sample Preparation for LC-MS^2^

For protein extraction, NCH644 GSCs were treated with 50 nM calcitriol one day after seeding. After 48 h, the cells were lysed in 2% SDS, 50 mM Tris-HCl pH8, 150 mM NaCl, 10 mM TCEP, 40 mM chloracetamide, protease inhibitor cocktail tablet (Sigma-Aldrich, Darmstadt, Germany) and “PhosStop” Phosphatase inhibitor tablet (Roche, Grenzach-Wyhlen, Germany), followed by sonication (1 ON/OFF, 30 s, 40%). Proteins were precipitated using methanol-chloroform extraction [69]. Proteins were digested with LysC (Wako Chemicals) and Trypsin (Promega, V5113), with a final ratio of 1:100, and digestion was performed overnight at 37 °C and stopped by trifluoroaceticacid (TFA). The peptides were purified using SepPak C18 columns (Waters, WAT054955). Eluates were dried and labeled with TMT reagents (ThermoFisher Scientific, 90061, TH266884) in a 1:2 (*w*/*w*) ratio in 50 mM TEAB (SIGMA, 86600) with 20% acetonitrile. The reaction was quenched with hydroxylamine to a final concentration of 0.5% at RT and samples were pooled in equimolar ratio. Finally, the peptides were fractionated using a High pH Reversed phase fractionation kit (ThermoFisher Scientific), according to the manufacturer’s instructions.

#### 4.6.2. Mass Spectrometry

Peptides were resuspended in 3% acetonitrile and 0.1% formic acid and separated on an Easy nLC 1200 (ThermoFisher Scientific) and a 22 cm long, 75 μm ID fused-silica column, which had been packed in house with 1.9 μm C18 particles (ReproSil-Pur, Dr. Maisch) and kept at 45 °C using an integrated column oven (Sonation). Peptides were eluted by a non-linear gradient from 5 to 60% acetonitrile over 155 min and directly sprayed into an Orbitrap Fusion Lumos mass spectrometer equipped with a nanoFlex ion source (ThermoFisher Scientific) at a spray voltage of 2.3 kV [70,71]. For analysis, 1/10 of each fraction was loaded onto the column. Each analysis used the Multi-Notch MS3-based TMT method [72] to reduce ion interference compared to MS2 quantification [73]. The scan sequence began with an MS1 spectrum (Orbitrap analysis; resolution 120,000 at 200 Th; mass range, 350–1400 m/z; automatic gain control (AGC) target, 400,000; normalized AGC target, 100%; maximum injection time, 100 ms). Precursors for MS2 analysis were selected using 10 ms activation time method. MS2 analysis consisted of collision-induced dissociation (quadrupole ion trap analysis; turbo scan rate; AGC 15,000; normalized AGC target, 150%; isolation window, 0.7 Th; normalized collision energy (NCE), 35; maximum injection time, 50 ms). Monoisotopic peak assignment was used, and previously interrogated precursors were excluded using a dynamic window (150 s ± 7 ppm); dependent scans were performed on a single charge state per precursor. Following acquisition of each MS2 spectrum, a synchronous-precursor-selection (SPS) MS3 scan was collected on the top 10 most intense ions in the MS2 spectrum [72]. MS3 precursors were fragmented by high energy collision-induced dissociation (HCD) and analyzed using the Orbitrap (NCE, 65; AGC, 100,000; normalized AGC target, 200%; maximum injection time, 150 ms; resolution was 15,000 at 200 Th).

#### 4.6.3. Data Analysis

Raw data were analyzed with Proteome Discoverer (PD) 2.4 (ThermoFisher Scientific), and SequenceHT node was selected for database searches. Human trypsin digested proteome (Homo sapiens SwissProt database (TaxID:9606, version 12 March 2020)) was used for protein identifications. Contaminants (MaxQuant “contamination.fasta”) were determined for quality control. TMT6 (+229.163) at the N-terminus, TMT6 (K, +229.163) at lysine and carbamidomethyl (+57.021), and cysteine residues were set as fixed modifications. Methionine oxidation (M, +15.995) and acetylation (+42.011) at the protein N-terminus were set for dynamic modifications. Precursor mass tolerance was set to 7 ppm and fragment mass tolerance was set to 0.5 Da. Default percolator settings in PD were used to filter perfect spectrum matches (PSMs). Reporter ion quantification was achieved with default settings in consensus workflow. Protein file from PD was then exported to Excel for further processing. Normalized abundances from protein file were used for statistical analysis after contaminations and complete empty values were removed. Significantly altered proteins were determined by a two-sided, unpaired Student’s *t*-tests (*p*-value < 0.05), adding minimum fold-change cut-off (≥0.5) with R version 4.0.2 [74] in RStudio [75]. Gplots version 3.1.1 [76] was used to visualize heatmaps, and EnhancedVolcano version 1.6.0 [77] was used for volcano plot. Figures were later edited with Adobe Illustrator CS5. The mass spectrometry proteomics data have been deposited to the ProteomeXchange Consortium via the PRIDE [78] partner repository with the dataset identifier PXD026789.

### 4.7. Lentiviral Transduction of GSCs

GFP-positive NCH644 cells were created by using pLV[Exp]-EGFP/Puro-CMV>Stuffer300 (Vectorbuilder GmbH, Neu-Isenburg, Germany). In total, 150,000 HEK293T cells were seeded in 6-well plates and allowed to incubate overnight. For transfection, 2 μg plasmid DNA (pLV(Exp)), 1.5 μg gag/pol plasmid (psPAX2, addgene #12260) and 0.5 µg VSV-G envelope plasmid (pMD2.G, addgene #12259) were used in 57 µL Opti-MEM and 6 µL FuGENE HD (Promega, Fitchburg, WI, USA) transfection reagent. Six hours later, the medium was changed, while the viral supernatant was collected after 16h and 40, respectively, and later pooled, as well as filtered through a 0.45 µm filter. The viral supernatant was diluted 1:1 with medium and 8 µg/mL protamine sulfate in PBS was added (Sigma-Aldrich). In total, 30,000 freshly dissociated cells were incubated for at least 48 h in virus-containing medium. Selection of positively transduced cells was ensured by adding 2 µg/mL or 1 µg/mL puromycin to culture medium for NCH644 and NCH421k, respectively. Transduction efficiency was determined by FACS measurement and was 91.6% for NCH644 GFP-positive cells (NCH644 GFP^+^) and 81.91% for GFP-positive NCH421k cells (NCH421k GFP^+^). psPAX2 was a gift from Didier Trono (Addgene plasmid # 12260; http://n2t.net/addgene:12260; RRID:Addgene_12260; accessed on 22 March 2021). pMD2.G was a gift from Didier Trono (Addgene plasmid # 12259; http://n2t.net/addgene:12259; RRID:Addgene_12259; accessed on 22 March 2021).

### 4.8. Adult Organotypic Slice Cultures and Ex Vivo Tumor Growth Assay

Adult organotypic tissue slice culture (OTC) was carried out based on the method described previously [29,41]. Mouse brains were dissected and dura mater was removed. Subsequently, mouse brains were placed in warm (35–40 °C) 2% low-melting agarose (Carl Roth). A Vibratome VT1000 (Leica, Wetzlar, Germany) was used to create evenly sized 150 µm transverse sections. The sections were placed on Millicell cell culture inserts (Merck KGaA, Darmstadt, Germany) and cultured in 6-well plates using FCS-free medium consisting of DMEM/F12 supplied with 1 × B27, 1 × N2 supplement and P/S (all from Gibco). One day later, multiple spheres were placed on the mouse brain slices. Adequate spheres were prepared by seeding 3000 NCH644 GFP-positive cells/well in u-shaped 96-well plates using 200 µL medium. Spheres were allowed to grow for 3 days. One day after sphere transplantation, pictures were taken (day 0) and the treatment was started, which was refreshed 3 times per week. Tumor growth was evaluated using FIJI, after pictures were taken regularly by a Nikon SMZ25 stereomicroscope equipped with a P2-SHR Plan Apo 2 × objective operated by NIS elements software. As the tumor size was normalized to the size on day 0, growth curves were created.

### 4.9. Statistics

Statistical analyses involved one-way and two-way ANOVA using GraphPad Prism 7 (GraphPad Prism 7 (GraphPad Software, La Jolla, CA, USA)), with the respective post hoc test, as indicated. For LDA, the statistical evaluation was taken from ELDA software, which calculated statistical significance based on a chi^2^-square test [31].

## 5. Conclusions

Here, we show that therapeutic doses of the hormonally active form of VitD_3_, calcitriol, carry potential as a means of anti-glioblastoma therapy. We demonstrated that calcitriol reduces the sphere-forming potential of GSCs in vitro, which was accompanied by reduced stemness marker gene expression and increased differentiation marker expression. Most importantly, using adult OTCs as a physiological and complex model system, we show that calcitriol not only hinders tumor growth as a single agent but also potently synergizes with the current standard-of-care TMZ. These promising findings emphasize that future research should be focused on further delineating the effects of calcitriol, a safe and well-known drug, and its application as an adjuvant therapy for GBM.

## Figures and Tables

**Figure 1 cancers-13-03577-f001:**
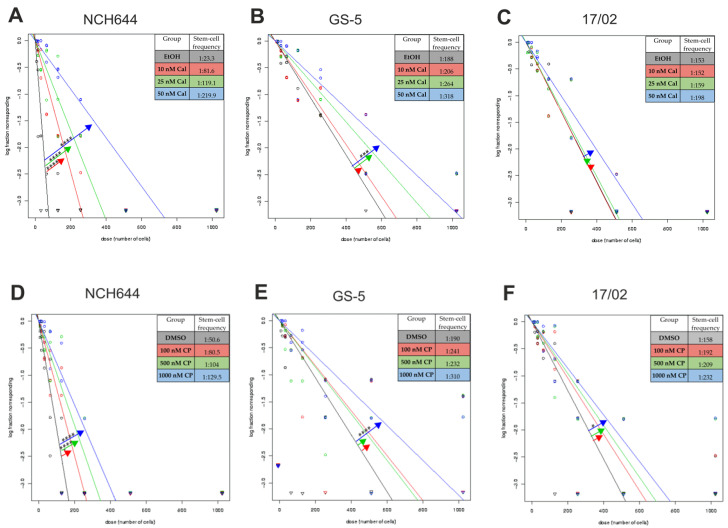
Calcitriol exerts differential activity among GSCs: (**A**–**F**) A log-fraction plot of the limited dilution model of data from (**A**,**D**) NCH644, (**B**,**E**) GS-5 and (**C**,**F**) 17/02 GSCs. The cells were treated with solvent (ethanol, EtOH; black) or (**A**–**C**) 10 (red), 25 (green) or 50 nM of calcitriol (blue) or (**D**,**F**) 100 (red), 500 (green) or 1000 nM of calcipotriol for 7 days after seeding the cell in a dilution series from 1024 to 8 cells and analyzing the data using ELDA software [31]. The estimated stem-cell frequency is presented under each plot. The data are the summary of at least three experiments performed with 12 replicates per cell number and 8 different cell numbers seeded. * *p* < 0.05; *** *p* < 0.001; **** *p* < 0.0001; chi-square-test from ELDA Web-App [31].

**Figure 2 cancers-13-03577-f002:**
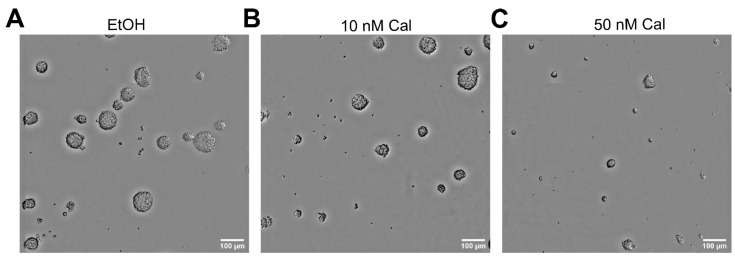
Calcitriol reduces NCH644 sphere size and number. Photomicrographs of NCH644 GSCs 7 days after treatment with (**A**) solvent (ethanol, EtOH), (**B**) 10 nM of calcitriol (Cal) or (**C**) 50 nM of Cal. Pictures were taken with a Tecan Spark plate-reader and images were cropped using FIJI (v1.52p) [33].

**Figure 3 cancers-13-03577-f003:**
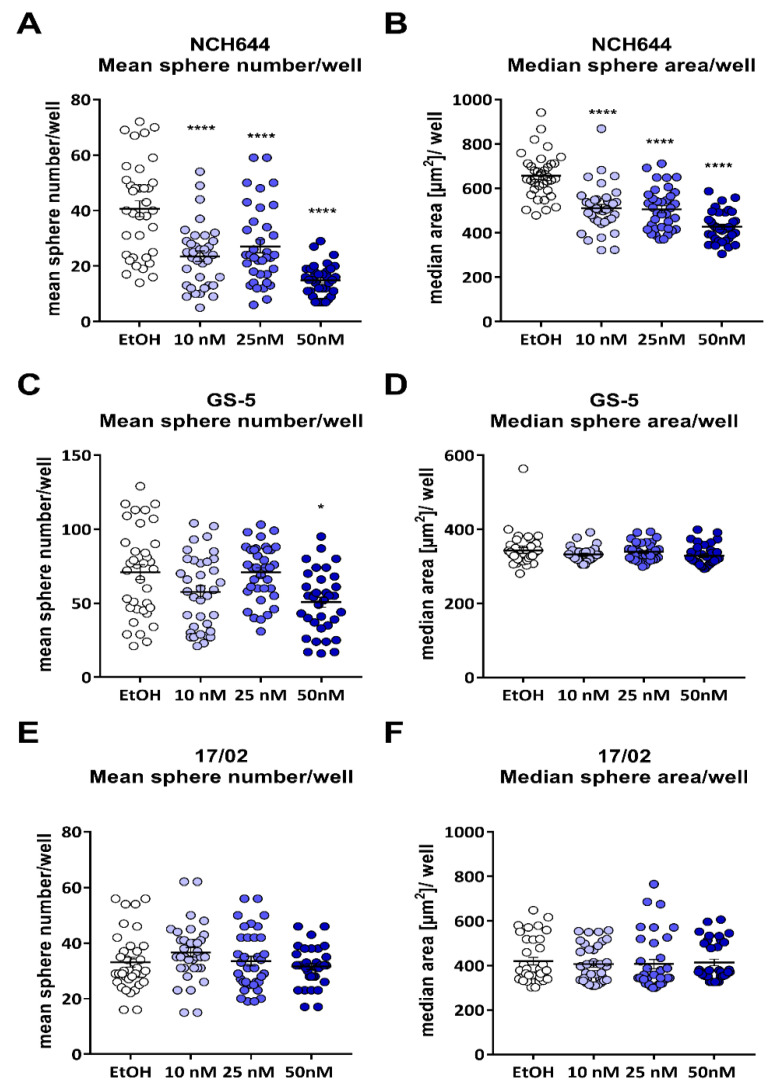
Calcitriol reduces sphere number and sizes of NCH644 and, to a lesser extent, in GS-5 GSCs: (**A**,**B**) Point-plots of NCH644 GSCs 7 days after treatment with solvent (ethanol, EtOH), 10 nM of calcitriol (Cal, light blue), 25 nM of Cal (blue) or 50 nM of Cal (dark blue) after seeding of 512 cells after analyses using a self-made FIJI-macro to measure (**A**) sphere number, (**B**) median sphere size. (**C**,**D**) Point-plots of GS-5 GSCs 7 days after seeding of 1024 and treatment as in (**A**). (**E**,**F**) Point-plots of 17/02 primary GSCs 7 days after seeding of 512 cells as in (**A**). * *p* < 0.05; **** *p* < 0.0001; one-way ANOVA Dunnett’s multiple comparisons test.

**Figure 4 cancers-13-03577-f004:**
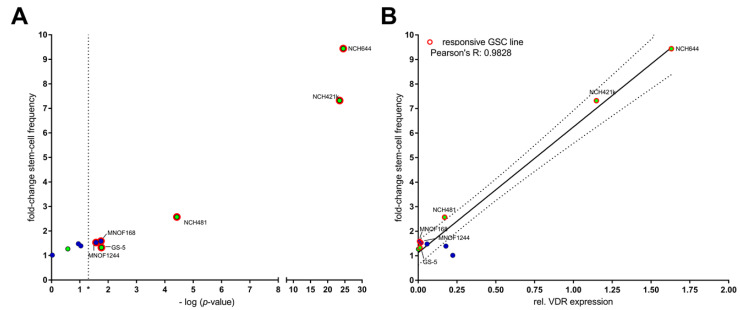
Differential sensitivity to calcitriol strongly correlates with VDR expression: (**A**) Point-plot of the relative fold-change in stem-cell frequency after performing a limiting dilution assay after treatment with 50 nM calcitriol or solvent for 7 days after seeding of 1024 to 8 cell per well. The stem-cell frequency was determined using ELDA [31] and the change in stem-cell frequency was calculated and plotted against the −log *p*-value (Chi^2^) of the ELDA-analysis. A *p*-value smaller than 0.05 was considered as statistically significant is marked by the asterisk (*) on the x-axis and the dotted, vertical line. (**B**) Point-plot of the relative fold-change in stem-cell frequency as in (**A**) plotted against the relative VDR expression and Pearson correlation (black line + 95% confidence interval; dotted line). Locally established primary cultures are depicted with blue dots, non-primary cultures are depicted with green dots. GSCs with a significant reduction in stem-cell frequency (right from the dotted line in (**A**)) are labeled and marked by red outlines in both sub-figures.

**Figure 5 cancers-13-03577-f005:**
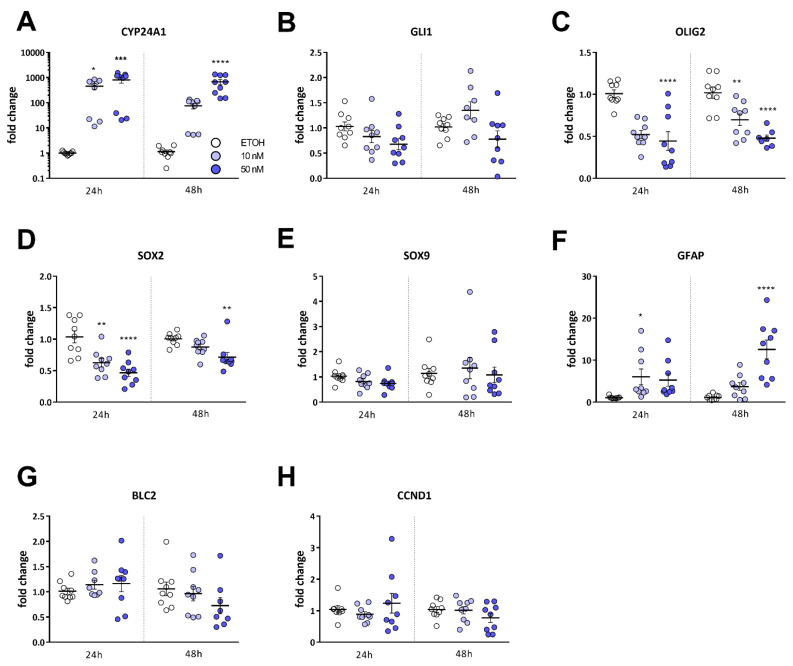
Calcitriol-induced changes in gene expression are indicative of reduced stemness. (**A–H**) Point-plots of Taqman-based gene expression of NCH644 GSCs after treatment with solvent (ethanol, EtOH), 10 (light blue) or 50 (dark blue) nM of calcitriol (Cal) for 24 or 48 h and measurement of (**A**) CYP24A1, (**B**) GLI1, (**C**) OLIG2, (**D**) SOX2, (**E**) SOX9, (**F**) GFAP, (**G**) BCL2 and (**H**) CCND1 expression. The data are the summary of three experiments performed in triplicates. * *p* < 0.05; ** *p* < 0.01; *** *p* < 0.001; **** *p* < 0.0001; one-way ANOVA with Dunnett’s multiple comparisons test.

**Figure 6 cancers-13-03577-f006:**
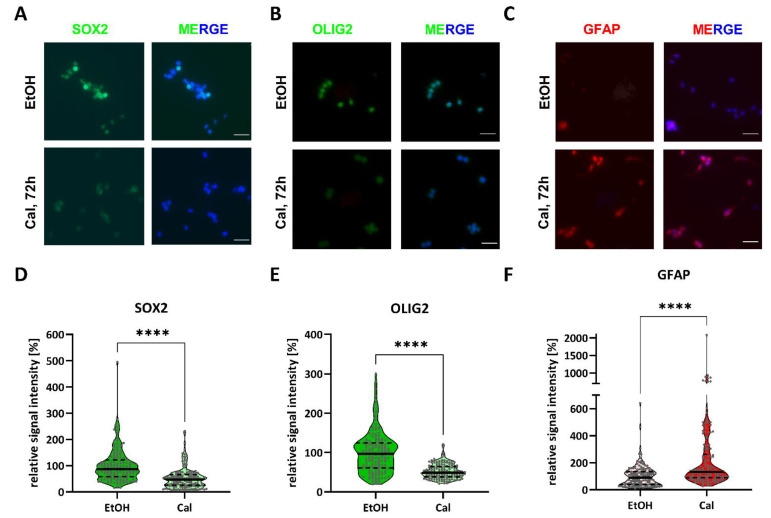
Calcitriol reduces SOX2 and OLIG2 stemness marker expression and increases GFAP differentiation marker expression: (**A**–**C**) representative immunofluorescence pictures of NCH644 GSCs 72 h after treatment with 50 nM of calcitriol (Cal) or solvent (EtOH) and (**D**–**F**) violin plots of the quantification of 50 cells from at least three random vision fields taken at 20× magnification of three independent experiments after staining against (**A**,**D**) SOX2, (**B**,**E**) OLIG2 and (**C**,**F**) GFAP. The solid horizontal line marks the median and the dashed lines separate the quartiles. **** *p* < 0.0001; two-tailed *t*-test. Scale bar: 50 µm.

**Figure 7 cancers-13-03577-f007:**
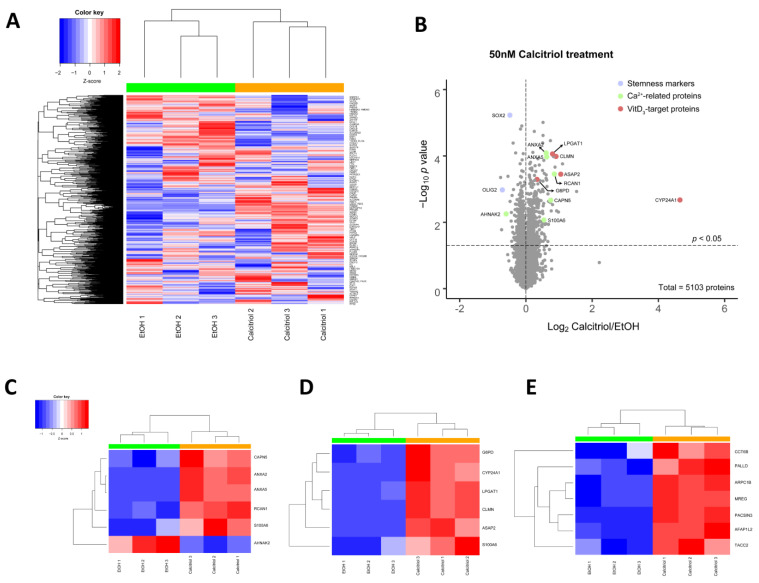
The effect of 48 h of calcitriol treatment on global protein levels: (**A**) hierarchal Euclidean clustered whole proteome indicates calcitriol similarly grouped and distinct profile of proteins. (**B**) Volcano plot showing fold-changes in calcitriol versus solvent for all quantified proteins. The stemness markers, Ca^2+^-related and vitamin D_3_-target proteins are also indicated and color coded on the plot. The heatmap represents Ca^2+^-related (**C**), vitamin D_3_-target (**D**) and actin/microtubules-related proteins (**E**).

**Figure 8 cancers-13-03577-f008:**
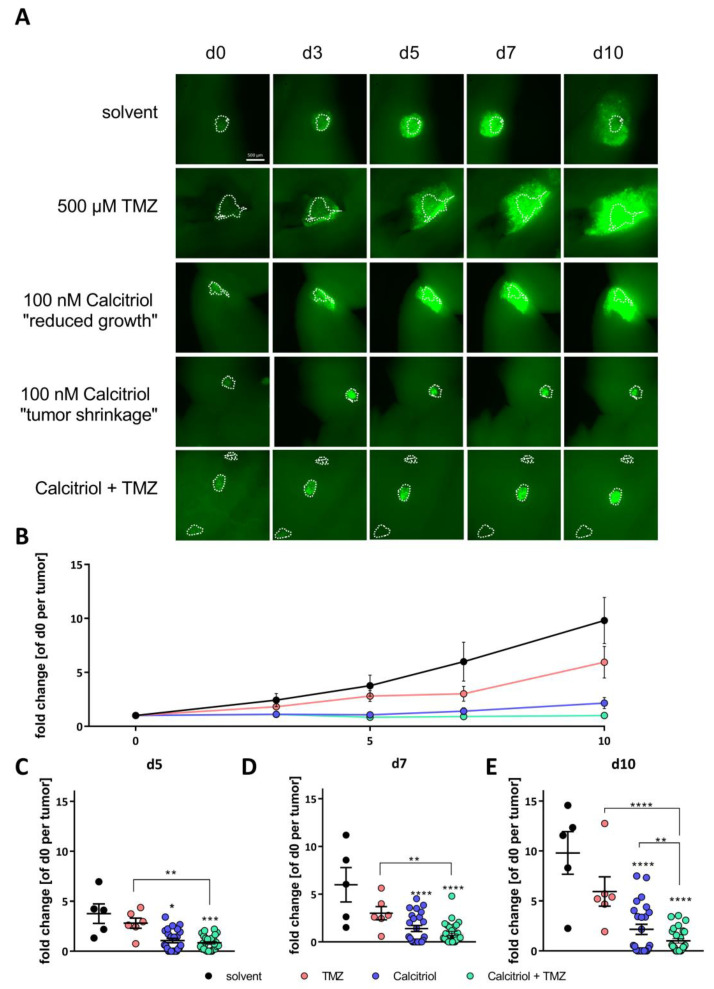
Calcitriol halts tumor growth and synergizes with TMZ in ex vivo organotypic tissue culture tumor growth kinetics: (**A**) representative microphotographs of NCH644 GFP^+^ tumors transplanted onto adult murine brain organotypic tissue culture slices after treatment with solvent (DMSO for TMZ; EtOH for calcitriol), 500 µM of TMZ, 100 nM of calcitriol (Cal) or a combination of both for the times indicated; scale bar: 500 µm. (**B**) Growth curves of the NCH644 GFP^+^ tumors normalized to the size of tumor d0 after treatment as in (**A**) depicted as the mean +/− SEM. (**C**–**E**) Point-plot of the data summarized in (**B**) after treatment for (**C**) 5 days, (**D**), 7 days and (**E**) 10 days, as in (**A**). * *p* < 0.05; ** *p* < 0.01; *** *p* < 0.001; **** *p* < 0.0001; two-way ANOVA with Tukey’s multiple comparisons test.

**Figure 9 cancers-13-03577-f009:**
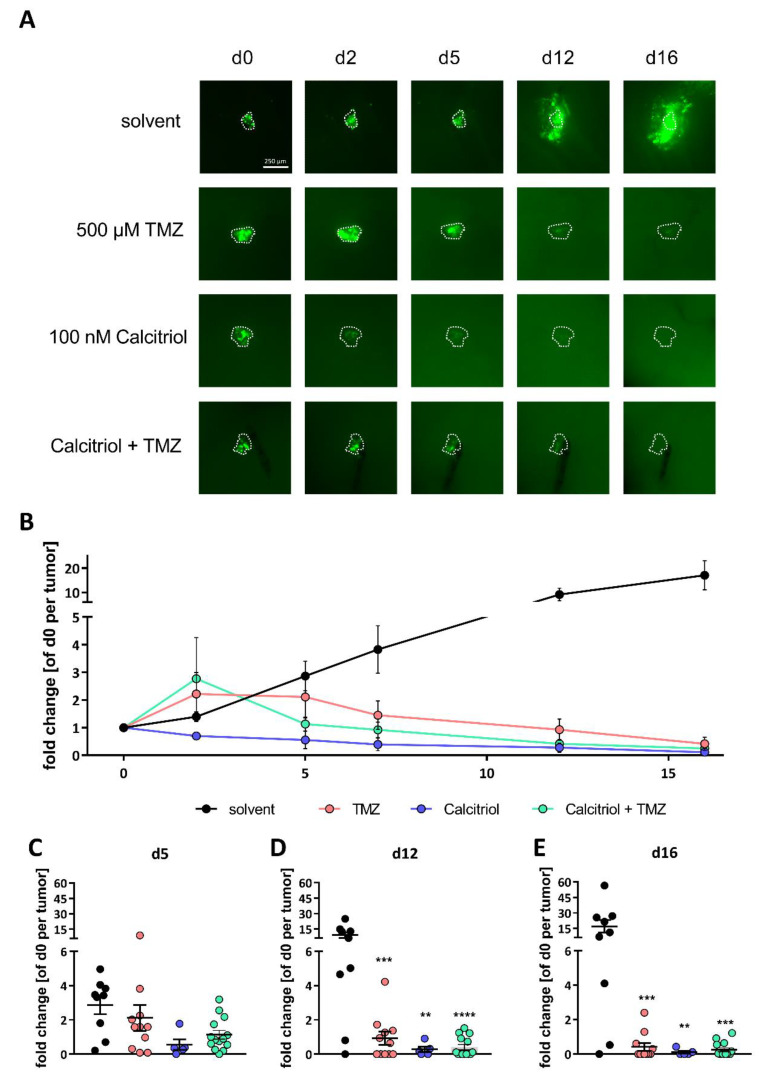
Calcitriol and TMZ prevent tumor growth in ex vivo organotypic tissue culture tumor growth kinetics: (**A**) representative microphotographs of NCH421k GFP^+^ tumors transplanted onto adult murine brain organotypic tissue culture slices after treatment with solvent (DMSO for TMZ; EtOH for calcitriol), 500 µM of TMZ, 100 nM of calcitriol (Cal) or a combination of both for the times indicated; scale bar: 250 µm. (**B**) Growth curves of the NCH421k GFP^+^ tumors normalized to the size of tumor d0 after treatment as in (**A**) depicted as the mean ± SEM. (**C**–**E**) Point-plot of the data summarized in (**B**) after treatment for (**C**) 5 days, (**D**), 12 days and (**E**) 16 days, as in (**A**). ** *p* < 0.01; *** *p* < 0.001; **** *p* < 0.0001; two-way ANOVA with Tukey’s multiple comparisons test.

**Table 1 cancers-13-03577-t001:** Calculated stem-cell frequencies, as determined using ELDA [31], of the GSCs NCH644, GS-5 and 17/02, 7 days after treatment with calcitriol, calcipotriol or solvent (EtOH, DMSO, respectively), as indicated.

Cell Line	Calcitriol [nM]	Stem-Cell Frequency [1/X]	*p*-Value (Chi^2^)	Calcipotriol [nM]	Stem-Cell Frequency [1/X]	*p*-Value (Chi^2^)
NCH644	EtOH	23.3		DMSO	50.6	
	10	81.6	1.46 × 10^−15^	100	80.5	0.00314
	25	119.1	3.01 ×10^−25^	500	104	5.05 × 10^−6^
	50	219.9	1.12 × 10^−43^	1000	129.5	4.58 × 10^−9^
GS-5	EtOH	188		DMSO	190	
	10	206	0.525	100	241	0.0861
	25	264	0.0245	500	232	0.140
	50	318	0.00061	1000	310	0.000556
17/02	EtOH	153		DMSO	159	
	10	152	0.97	100	192	0.191
	25	159	0.81	500	209	0.0627
	50	198	0.101	1000	232	0.0109

## Data Availability

The mass spectrometry proteomics data have been deposited to the ProteomeXchange Consortium via the PRIDE [78] partner repository with the dataset identifier PXD026789. Additional raw data can be made available upon reasonable request.

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
