# Peer review of "Calcitriol Promotes Differentiation of Glioma Stem-Like Cells and Increases Their Susceptibility to Temozolomide"

_cancers, 2021, doi:10.3390/cancers13143577_

Round 1

Reviewer 1 Report

The huge efforts made to address comments improved the quality of the manuscript that is now suitable to be accepted by Cancers journal.

Reviewer 2 Report

In the revised manuscript, the authors have addressed most of the concerns and they have answered many of our initial queries, so the main scientific message is stronger and clearer in this version.

In particular the authors have included the analysis of calcitriol efficacy in 7 additional GSC cultures; moreover, they have performed proteomic expression analyses (liquid chromatography and tandem mass spectrometry) in the highest responding cell line NCH644 and argued these new data  in the revised manuscript.

I think that this new version of the manuscript  “Calcitriol promotes differentiation of glioma stem-like cells and increases their susceptibility to temozolomide” it will be acceptable for the publication in Cancers.

This manuscript is a resubmission of an earlier submission. The following is a list of the peer review reports and author responses from that submission.

Round 1

Reviewer 1 Report

In the current manuscript entitled “Calcitriol promotes differentiation of glioma stem-like cells and 2 increases their susceptibility to temozolomide” the authors demonstrated that the calcitriol can differently reduce stemness in three GSC lines and tumor growth in GSCs ex vivo model; and to shows a synergistic effect when combined with Temozolomide (TMZ) further reducing the GSC growth in ex vivo model. The authors provided evidence assessing in vitro studies on NCH644, GS-5, and 17/02 cell lines. Among these, only one cell culture, 17/02, has been defined as human primary cell culture.

Although the purpose of the study is very interesting and could set the basis for further treatment options associable with the gold standard of this malignancy, the manuscript is not well written, lacks details, and is weak in some points. An English native speaker review is recommended, indeed, the sentences could be better structured for a higher and faster understanding of the text.

For this reason, a major revision is needed to improve the manuscript not acceptable in this form.  

Abstract: The authors should include and stress the objective of the study in this section.

Introduction: The authors introduced vitamin D synthesis and metabolism, however, several parts of the description are quite weak, and in some instances misleading or incorrect. Indeed, CYP27A1, widely distributed in several tissues with highest levels in liver and muscle, but also in kidney, intestine, lung, skin and bone, may have 25-hydroxylase activity, but there is now excellent genetic evidence in mice and in humans that CYP2R1 is the major 25-hydroxylase. In the kidney the 1a hydroxylation occurs by CYP27B1. (Reference: DOI: 10.3390/nu12061798). Please completely rephrase the sentences starting from line 60 to 64, introducing CYP27B1 that has been cited in line 71.

Line 43:  I would suggest to include a high-impact reference referred to Tumors- Treating Fields, doi.org/10.1038/s41416-020-01136-5.

Line 55-77: The paragraph needs to be restructured and rewritten. Firstly, the authors should describe vitamin D synthesis and degradation, followed by VDR activation, and then the implication of acidosis on 24 hydroxylase activity.

Line 78: The authors stated, “It was recently shown that serum level of 25(OH)D3 and the risk of GBM have an inverse relationship while VDR-expression in these tumors was found to be associated with improved overall survival”. Do the authors mean that 25(OH)D3 serum levels are inversely correlated with the risk to develop GBM? Please use a more scientific language.

The authors should mention at the end of the introduction the aim of their research, avoiding to anticipate the results.

Results: The flow of this section is not fluid. I suggest you divide it into different paragraphs in which the authors should merely describe the obtained results avoiding to include details that should be reported or discuss with other sections such as “methodology or discussion”. The results should report all treatment times, drug concentrations used, percentage and statistical significance etc. The discussion of the results obtained supported with other scientific evidence must be reported in the discussion.

The spelled-out form of each acronym should be indicated on first use (e.g. TMZ+IR).

Pay attention to the figure numbers reported in the text. Lines 158 and 159 should be reported fig 4 and 4A; lines 189 and 191 should be reported fig 5A and 5B.

Discussion: The discussion should be better implemented and integrated with the results of the study.

Materials and Methods: This section lacks details, it’s fine to cite own previous studies but the experiment setting is mandatory. It could be helpful if the authors include a study design in which they explain the choice of drug concentration, time exposure, the rationale of the study, etc. E.g: how many days after the cell-seeded the authors have treated the cells with calcitriol? Please, the authors should include the drug concentration used.

The author stated “17/02 is a primary cell line obtained from a 60 year old patient” without reporting any reference. Therefore, I’m wondering if the authors had an approved consent by an ethic committee and a signed informed consent from the patient.

Please, provide details of limiting dilution assay, including drug concentration used and time of exposure, and of ELDA analysis.

Further experiments:

The author demonstrated that only one cell line, NCH644, shows to response to calcitriol treatment in terms of decreased sphere number and smaller sphere size of GSC, although after 24 and 48 hrs of treatment the CYP24A1 was significantly increased. How do the authors justify this result? Do the authors think this effect is caused by different VDR expression levels in the three cell lines? The authors should perform at list a RT-qPCR to evaluate the VDR expression at mRNA level.

Minor revision

Figure 1. Please, the authors should include in the graphs the statistical significance and p value.

Figure2. Please include what magnification the photos were taken.

Line 153: Please, move this sentence to the methodology section.

Reviewer 2 Report

In this study the authors described how the effects of therapeutic doses of calcitriol  in GSC lines suppress stemness and that calcitriol synergizes with temozolomide ex vivo to completely eliminate some GSC tumors. The driving idea of this work is very interesting as new knowledge regarding drugs increasing temozolomide therapy is needed.  

However, in order to keep the conclusion regarding calcitriol efficacy, validation in a larger panel of GSCs are necessary. The authors showed results  by in vitro limiting dilution assay, obtained in 3 GSC lines, and only in one the reduction of stemness is strong.

Further, the authors provided evidence of gene expression changes induced by calcitriol-treatment after 24-48hs by qRT-PCR in one GSC line. Gene expression analysis should be implemented by a gene-array analysis.